# Associations between COVID-19 Work-Related Stressors and Posttraumatic Stress Symptoms among Chinese Doctors and Nurses: Application of Stress-Coping Theory

**DOI:** 10.3390/ijerph19106201

**Published:** 2022-05-19

**Authors:** Rui She, Lijuan Li, Qian Yang, Jianyan Lin, Xiaoli Ye, Suliu Wu, Zhenggui Yang, Suzhen Guan, Jianxin Zhang, Rachel Hau Yin Ling, Joseph Tak Fai Lau

**Affiliations:** 1Centre for Health Behaviours Research, JC School of Public Health and Primary Care, The Chinese University of Hong Kong, New Territory, Hong Kong, China; sherryshe0319@link.cuhk.edu.hk (R.S.); hauyinling@cuhk.edu.hk (R.H.Y.L.); 2School of Public Health, Dali University, Dali 671000, China; lelejuan@126.com; 3Center for Health Policy Studies, Department of Endocrinology, School of Public Health, Children’s Hospital, National Clinical Research Center for Child Health, Zhejiang University School of Medicine, Hangzhou 310003, China; chianyoung@zju.edu.cn; 4The Fourth People’s Hospital of Nanning, Nanning 530023, China; linjianyan@126.com; 5The Children’s Hospital Zhejiang University School of Medicine, Hangzhou 310003, China; yexiaoli@zju.edu.cn; 6Wuyi First People’s Hospital, Jinhua 321200, China; 8015059@zju.edu.cn; 7No. 4 Hospital of Ningxia Hui Autonomous Region, Yinchuan 750021, China; yangzhenggui_1983@126.com; 8Key Laboratory of Environmental Factors and Chronic Disease Control, School of Public Health and Management, Ningxia Medical University, Yinchuan 750004, China; guansz_nx2017@sina.com; 9Huaxi School of Public Health, Sichuan University, Chengdu 610041, China; zhangjianxin955@163.com

**Keywords:** COVID-19, healthcare workers, posttraumatic stress symptoms, stress-coping theory, China

## Abstract

Healthcare workers are vulnerable to posttraumatic stress symptoms (PTSS) due to stressful work during the COVID-19 pandemic. This study aimed to investigate whether the associations between COVID-19 work-related stressors and PTSS would be mediated by maladaptive and adaptive coping strategies and moderated by perceived family support based on stress-coping theory. An anonymous online survey was conducted among 1449 doctors and nurses in five hospitals in China between October and November 2020 during the “post-outbreak” period. The prevalence of PTSS assessed by the Posttraumatic Symptom Scale Self-Report was 42%. Logistic regression analysis revealed that worries about being infected with COVID-19, perceived difficulties in family caregiving, coping strategies of rumination, catastrophizing, acceptance, and perceived family support were independently associated with PTSS. Furthermore, maladaptive and adaptive coping partially mediated the association between COVID-19 work-related stressors and PTSS.The results of multi-group analyses showed that perceived family support tended to intensify the associations between COVID-19 work-related stressors and adaptive coping and between adaptive coping and PTSS, whereas perceived family support attenuated the positive association between COVID-19 work-related stressors and PTSS. The findings suggest tailor-made health interventions with respect to alleviation of work-related stressors and coping skill training to reduce the risk of PTSS among healthcare workers, especially for those with lower perceived family support.

## 1. Introduction

The coronavirus disease (COVID-19) pandemic has substantially diminished people’s social, economic, and psychological resources worldwide. Mental distress due to the COVID-19 pandemic is prevalent, especially for healthcare workers (HCWs) [1]. Common work-related stressors include shortage of medical resources, overwhelming workload, high risk of infection, fatigue, and perceived inability to treat patients, all of which are potentially traumatic to HCWs [2]. Acute stress reactions (ASR) and posttraumatic stress symptoms (PTSS) related to the pandemic have been reported. The former refers to the initial symptoms arising very soon after being exposed to a traumatic event, whereas the latter is characterized by recurrent memories, avoidance, and heightened arousal regarding a traumatic event that lasts long after its occurrence [3]. Individuals commonly develop PTSS within three months of the trauma, but its symptoms may appear later and often persist for months and sometimes years [3].

Prior literature has reported that experiences encountered during outbreaks of emerging infectious diseases [e.g., severe acute respiratory syndrome and Middle East respiratory syndrome] were associated with PTSS among HCWs [4]. A recent systematic review of nine studies also reported a pooled prevalence of PTSS of 21.5% among HCWs during the COVID-19 pandemic [5]. Nevertheless, most of these studies assessed the prevalence of PTSS during or shortly after the COVID-19 outbreak e.g., during February to March 2021 among Chinese HCWs [6] and during March to April 2021 among Italian HCWs [2], i.e., when the traumatic conditions were still ongoing. Such assessments may be unable to distinguish between ASR and PTSS. To monitor the longer-term impact of COVID-19, it is important to investigate PTSS among HCWs during a “post-outbreak” period, i.e., when the newly detected COVID-19 cases have completely or largely subsided. Under those circumstances, individuals showing PTSS may develop lasting problems and would require interventions [7]. China is one of the few countries that offers such a unique context for investigating PTSS among HCW during a “post-outbreak” period. Whereas many countries are still reporting a large number of new cases, the national daily number of newly reported COVID-19 cases in China peaked at 15,512 on 12 February 2020 and then declined onward and dropped to less than 10 per day on 22 April 2020, remaining at a very low level ever since [8]. The present study was conducted from October to November 2020, which was about six months since COVID-19 was “put under control” in China, when hospitals in China were providing normal services to the public [9]. The assessment of PTSS among HCWs during that time period was appropriate.

Multiple factors were associated with PTSS regarding COVID-19 among HCWs, among which work-related stressors (e.g., heavy workload, working in unsafe settings, and lack of training) were especially relevant [2]. For instance, previous studies have suggested that HCWs providing direct care to COVID-19 patients were at higher risk of developing PTSS than others [10]. Worries and fear about their and their colleagues’ risk of infection were stressful and positively associated with PTSS [11]. With the overwhelming workload and requirement to stay overnight within hospitals, HCWs reported difficulties taking care of their families during the COVID-19 outbreak and worries about infecting their families [12], and might have hence developed guilty feelings that would potentially lead to PTSS. The lack of confidence in handling future COVID-19 outbreaks may also negatively impact HCWs’ mental health and result in PTSS; a prior study during the H1N1 pandemic showed that perceived confidence in the local government’s ability to control future outbreaks was negatively associated with mental distress among Chinese university students [13]. These aforementioned factors of PTSS are important in designing programs preventing PTSS among HCWs but have rarely been investigated.

Identification of the underlying mechanisms between COVID-19 work-related stressors and PTSS is crucial to inform future interventions. The present study hypothesized that maladaptive and adaptive coping strategies would mediate the association between work-related stressors due to COVID-19 and PTSS among HCWs. Stressful experiences may lead to feelings of powerlessness, helplessness, and confusion, which might lead to a predisposition toward heightened maladaptive coping and lowered adaptive coping [14,15]. In addition, according to the stress-coping theory, how people evaluate and respond to stressors would affect their mental health outcomes [16]. Adaptive coping strategies (e.g., cognitive restructuring) focus on reducing the negative effects of the stressors and are associated with positive mental health outcomes [17]. In contrast, maladaptive coping such as rumination and catastrophizing thoughts, which is commonly triggered during a crisis (e.g., COVID-19), was positively associated with PTSS [18]. Empirically, recent studies found that adaptive coping mediated the association between COVID-19-related stressful experiences and ASR among Chinese college students [19] and coping strategies significantly mediated the association between stress due to COVID-19 work and secondary trauma among Italian emergency workers [20]. Adaptive and maladaptive coping strategies are thus plausible mediators between COVID-19 work-related stressors and PTSS.

Not all HCWs encountering stressful working experiences during the pandemic have developed PTSS. Understanding the moderation mechanisms affecting the association between COVID-19 work-related stressors and PTSS would facilitate the identification of HCWs who are particularly vulnerable to develop PTSS and improve the effectiveness of programs preventing PTSS. The moderation effect of perceived family support for the association between work-related stressors and PTSS was tested in this study. During the COVID-19 pandemic, family support is especially important in preventing mental illnesses and fostering recovery from traumas among HCWs [12]. Perceived support is a coping resource that may alter the cognitive appraisal and increase the possibility of feeling more in control of a stressful situation [21]. The stress-coping theory posits that the availability of coping resources would facilitate the use of adaptive coping strategies to deal with stressors and increase the efficacy of coping strategies [21,22]. Empirically, perceived social support buffered the harm of negative life events on psychological health among youth and adults [21,23]. Family support and active coping also attenuated the adverse impact of acculturative stress on mental health among college students [24]. It is plausible that perceived family support would synergistically amplify the protective effect of adaptive coping on PTSS while buffering the adverse impact of maladaptive coping on PTSS.

The present study investigated the level and associated factors of PTSS among Chinese HCWs during October to November 2021, which was about six months since the COVID-19 outbreak was “put under control” in China. The three categories of potential factors included: (a) COVID-19 work-related stressors, including engagement in frontline COVID-19 work, worries about infection via work, perceived difficulty in family caregiving, and perceived inability of oneself and the affiliated hospital in handling work during prospective future COVID-19 outbreaks, (b) adaptive and maladaptive coping strategies, and (c) perceived family support. It was hypothesized that maladaptive and adaptive coping strategies would mediate the association between COVID-19 work-related stressors and PTSS. Furthermore, perceived family support would moderate the paths of the mediation model between COVID-19 work-related stressors and PTSS. Specifically, it was hypothesized that the associations between COVID-19 work-related stressors and adaptive coping and between adaptive coping and PTSS would be stronger among HCWs with stronger perceived family support, while the associations of COVID-19 work-related stressors, maladaptive coping, and PTSS would be more evident among HCWs who perceived lower family support.

## 2. Methods

### 2.1. Sampling

An anonymous cross-sectional survey was conducted from October to November 2020. Five hospitals in four provinces (Zhejiang, Ningxia, Guangxi, and Yunnan) in mainland China were selected via existing collaboration network. The four provinces were geographically (east, north central, south, and southwest) and socioeconomically (levels of gross domestic product: top, about average, below average, and low) representative of mainland China to some extent. The inclusion criteria of the participants were: (a) full-time doctors or nurses, (b) employment in the current hospital since January 2020, and (c) access to mobile phones. All eligible doctors/nurses (*n* = 2419) working in the major departments of internal medicine, surgery, gynecology and obstetrics, pediatrics, emergency, infectious diseases, and intensive care were invited to complete an anonymous online survey. The online survey link was distributed to the prospective participants by the hospital administrators through the participating departments’ regular WeChat/QQ platforms, which were the most commonly used social media applications in China. All participants were briefed properly about the study. It was explained to them that the participation was voluntary and anonymous, and rejection would not cause any negative consequences. They were also guaranteed that only the research team could access their data. No incentives were given to the participants. The study was approved by the Survey and Behavioral Research Ethics Committee of the corresponding author’s affiliation (Reference No. SBRE-19-644). A total of 1449 completed questionnaires were returned to the research team; the response rate was 60.0% (1449/2419).

### 2.2. Measurements

#### 2.2.1. Background Variables

Data on socio-demographics (i.e., sex, age, marital status, and education level) and work-related variables (i.e., department, job seniority rank, profession, and hospital) were collected.

#### 2.2.2. Posttraumatic Stress Symptoms

The PTSD Symptom Scale-Self-Report (PSS-SR) was used to assess the level of PTSS related to COVID-19 during the past month, which was designed according to the DSM-IV criteria [25]. It comprises three subscales: avoidance, arousal, and re-experiencing. Sample items include “trying to avoid activities, people, or places that remind you of the illness” (avoidance) and “having trouble falling asleep or staying asleep” (arousal). Responses were rated on a 4-point scale (0 = not at all to 3 = almost always). Higher sum scores in the subscales indicated higher levels of PTSS. A score of 13 or higher indicates the likelihood of PTSS [26]. The scale was reliable and valid among Chinese cancer patients [27]. The Cronbach’s alphas of the three subscales were 0.90, 0.91, and 0.93, respectively.

#### 2.2.3. COVID-19 Work-Related Stressors

(1) Engagement in COVID-19 frontline work: Two original items were constructed to assess whether the participants had engaged in taking care of COVID-19 patients or clinical examinations of COVID-19 patients/suspected cases during the outbreak period. (2) Worries about being infected with COVID-19: Two items were used to assess the level of concerns about themselves and their colleagues being infected with COVID-19. Responses were rated on a 5-point Likert scale (1 = almost none to 5 = extremely high). Higher total scores indicated higher levels of worries about infection. (3) Perceived difficulty in family caregiving during the COVID-19 outbreak period: Two items were used to ask the participants whether they had difficulties or not in taking care of their children and older family members during the COVID-19 outbreak period. (4) Perceived inabilities of oneself and the affiliated hospital in handling work during prospective COVID-19 outbreaks in the next year: Two items were used to assess perceived abilities of oneself and the affiliated hospital, respectively. Responses were rated on a 5-point Likert scale (1 = strongly agree to 5 = strongly disagree). Higher scores denoted lower levels of perceived ability.

#### 2.2.4. Coping Strategies

Two maladaptive coping strategies (i.e., rumination and catastrophizing) were measured by the two subscales of the short version of the Cognitive Emotion Regulation Questionnaire (CERQ-short) [28]. Two adaptive coping styles (i.e., positive reframing and acceptance) were assessed by using two subscales of the Brief COPE [29]. These scales have been applied in a previous Chinese COVID-19 study [30]. Each subscale includes two questions [1 (almost never) to 5 (almost always)]. Sample statements include “I often think about how I feel about COVID-19”, “I keep thinking about how terrible COVID-19 is”, and “I think of something nice instead of what has happened”. In the present study, the Cronbach’s alphas of the four subscales were 0.66, 0.71, 0.81, and 0.82 for rumination, catastrophizing, acceptance, and positive reframing, respectively.

#### 2.2.5. Perceived Family Support

A single item was used to measure the level of perceived family support during the COVID-19 outbreak: “How much support had you received from your family during the COVID-19 outbreak?”. Responses ranged from 0 (none) to 10 (tremendous), which were further categorized into two groups of lower (score 0–7) and higher (score 8–10) levels for the multi-group analysis testing the moderation role of perceived family support. The two groups made up 25.7% and 74.3% (upper and lower quartiles) of the participants, respectively. A previous study has supported the validity and reliability of using single-item social support measurements [31].

### 2.3. Statistical Analysis

Descriptive statistics were presented. Multiple linear regression was performed to investigate the associations between the independent variables (i.e., background variables, COVID-19 work-related stressors, coping strategies, and perceived family support) and PTSS by entering all such factors into the same model. Standardized (*beta*) and unstandardized regression coefficients (B) and their corresponding 95% confidence interval (CI) are reported.

Structural equation modeling (SEM) analysis was used to test the hypothesized mediation model. Indicators of COVID-19 work-related stressors, maladaptive coping, adaptive coping, and PTSS were created by the item parceling method basing on the subscales or different dimensions of the construct [32]. Background variables (i.e., marital status) that were significantly associated with PTSS were controlled for. Several fit indices were used to assess the adequacy of model fit: (1) chi-square/degrees of freedom (χ^2^/df) ratio ≤ 3, (2) comparative fit index (CFI) ≥ 0.90, (3) incremental fit index (IFI) ≥ 0.90, (4) root mean square error of approximation (RMSEA) ≤ 0.08, and (5) standardized root mean square residual (SRMR) ≤ 0.08 [33,34]. The direct and indirect effects were estimated using bootstrapping, which is a non-parametric resampling procedure that involves repeated sampling of the dataset (*n* = 2000). The effect size (i.e., the proportion of mediation) was reported.

To examine the significance of each structural path between the lower and higher perceived family support groups, a multi-group SEM analysis was conducted. A series of models, each constraining a specific individual path (e.g., COVID-19 work-related stressors and adaptive coping strategies), were compared to the unconstrained model that freely estimated all the paths. In the analysis, *p* values < 0.05 in the chi-square difference test (Δχ^2^/Δdf) would denote a statistically significant moderation effect of perceived family support on that tested path. The SPSS 23.0 Statistics for Windows (IBM Corp. Released 2015, Armonk, NY, USA: IBM Corp) and AMOS 23.0 were used for all statistical analyses.

## 3. Results

### 3.1. Descriptive Statistics

The description of the participants is summarized in Table 1. The mean age of the participants was 34 years (standard deviation = 9.0 years). The majority were females (85.4%), nurses (70.8%), being married, and had obtained a bachelor’s degree or above. About half of them had engaged in COVID-19 frontline work (50.4%) and self-reported difficulties in taking care of their children or older family members during the COVID-19 outbreak period (53.9%). The mean scores (range: 2–10) of positive reframing (7.2) and acceptance (7.5) scales were higher than those of rumination (5.3) and catastrophizing (4.5). About three-fourths (74.3%) of the participants belonged to the group of higher levels of family support (score > 7).

The average total score of PTSS was 12.0 (95% CI: 11.5–12.5). Of the participants, 42.0% (95% CI: 39.5–44.6%) were classified as having PTSS. The differences in the mean PTSS scores (12.5 versus 11.8) and prevalence of PTSS (44.7% versus 40.9%) between doctors and nurses were statistically non-significant (*p* = 0.217 and *p* = 0.189, respectively).

### 3.2. Multiple Regression Analysis

The results of the multiple linear regression model are presented in Table 2. Five variables were statistically significantly and positively associated with PTSS, including (1) separated or divorced marital status (*beta* = 0.07, *p* = 0.005; reference group: single), (2) worries about being infected with COVID-19 (*beta* = 0.12; *p* < 0.001), (3) perceived difficulties in family caregiving (*beta* = 0.08; *p* = 0.001), (4) rumination (*beta* = 0.18; *p* < 0.001), and (5) catastrophizing (*beta* = 0.33; *p* < 0.001). Meanwhile, two factors were negatively associated with PTSS, i.e., the coping strategy of acceptance (*beta* = −0.15; *p* < 0.001) and perceived family support (*beta* = −0.07; *p* = 0.005). All the independent variables together explained 38.4% of the total variance of PTSS (*F* = 27.32, dfs = 31 and 1359; *p* < 0.001).

### 3.3. Correlation Analysis

The bivariate correlation analyses showed that all but two of the individual indicators of COVID-19 work-related stressors were statistically and significantly correlated with both the maladaptive coping and adaptive coping variables, with absolute values of r ranging from 0.05 to 0.40 (all *p* < 0.01). In addition, all the individual indicators related to COVID-19 work-related stressors, coping strategy variables, and perceived family support all showed statistically significant correlations with PTSS, with the absolute values of r ranging from 0.09 to 0.54 (all *p* < 0.01) (Table 3).

### 3.4. SEM Analysis

#### 3.4.1. Testing Mediation Effects

The structural model fitted the data well (χ^2^/df = 11.32, CFI = 0.91, IFI = 0.91, RMSEA = 0.08, SRMR = 0.07). All the parcel indicators were significantly loaded on the latent variables, with standardized factor loadings ranging from 0.14 to 0.98 (all *p* < 0.001). As shown in Figure 1, COVID-19 work-related stressors were positively associated with maladaptive coping (*beta* = 0.59; *p* < 0.001) and PTSS (*beta* = 0.33; *p* < 0.001), and negatively associated with adaptive coping (*beta* = −0.29; *p* < 0.001). Maladaptive coping was positively associated with PTSS (*beta* = 0.40; *p* < 0.001), whereas adaptive coping was negatively associated with PTSS (*beta* = −0.08; *p* = 0.002).

Regarding the indirect effects, the significant standardized partial mediation effect via maladaptive coping was 0.23 (*p* < 0.05) and the proportion mediated was 39.4%. The statistically significant and standardized mediation effect via adaptive coping was 0.03 (*p* < 0.05); however, only 4.1% of the association was explained by this indirect path. In addition, the direct effect of COVID-19-related stressors on PTSS was 0.33 (*p* < 0.05), which explained 55.8% of the total effect.

#### 3.4.2. Testing Moderation Effects

As shown in Table 4, the results of the multi-group SEM analysis indicated that perceived family support significantly moderated the paths (1) between COVID-19 work-related stressors and adaptive coping (Δχ^2^/Δdf = 96.94, *p* < 0.001), (2) between adaptive coping and PTSS (Δχ^2^/Δdf = 4.29, *p* = 0.038), and (3) between COVID-19 work-related stressors and PTSS (Δχ^2^/Δdf = 6.33, *p* = 0.012). The similar moderations for the association between COVID-19 work-related stressors and maladaptive coping (Δχ^2^/Δdf = 3.57, *p* = 0.059) and that between maladaptive coping and PTSS (Δχ^2^/Δdf = 3.21, *p* = 0.073) were statistically non-significant. To elaborate, higher levels of perceived family support would intensify the associations between COVID-19 work-related stressors and adaptive coping (*beta* = −0.23, *p* < 0.001 versus *beta* = −0.09, *p* = 0.061 in higher/lower perceived family support groups) and between adaptive coping and PTSS (*beta* = −0.12, *p* < 0.001 versus *beta* = −0.01, *p* = 0.628 in higher/lower perceived family support groups). In contrast, higher levels of perceived family support would attenuate the positive association between COVID-19 work-related stressors and PTSS (*beta* = 0.23, *p* < 0.001 versus *beta* = 0.46, *p* = 0.034 in higher/lower perceived family support groups).

## 4. Discussion

### 4.1. Interpretation of Principle Results

The present study is among the first ones investigating the relationship between various COVID-19 work-related stressors and PTSS among a large sample of HCWs, and detangling the potential underlying mechanisms of coping through the lens of stress-coping theory. About 40% of Chinese HCWs reported probable PTSS about six months since the COVID-19 outbreak was “put under control” in China. COVID-19 work-related stressors (i.e., worries about COVID-19 infection and perceived difficulties in family caregiving), coping strategies (i.e., rumination, catastrophizing, and acceptance), and perceived family support were significantly and independently associated with PTSS. Furthermore, maladaptive and adaptive coping strategies partially mediated the association between work-related stressors and PTSS. Notably, higher perceived family support intensified the “protective paths” between COVID-19 work-related stressors, adaptive coping, and PTSS, while buffering the “risky path” between COVID-19 work-related stressors and PTSS. Such findings have implications for designing nuanced and effective prevention programs of PTSS among HCWs.

The prevalence of PTSS in the current sample seemed higher than that recently reported in two other studies [6,35] and was similar to another recent study conducted among HCWs [36]. Variations in sample characteristics, sampling method, investigation time, and measurements may contribute to the observed differences across studies. For instance, most of the participants of the present study worked in hospitals designated to take care of COVID-19 patients; as a result, they might hence face stronger stress than their counterparts working in non-designated hospitals. Also, previous studies tended to measure PTSS in earlier periods [e.g., during February to March 2021 among Chinese HCWs [6], whereas PTSS sometimes takes time to develop and might be less observable during a traumatic period [7]. The trajectory of PTSS among HCWs and changes in prevalence over time needs follow-up research. As PTSS may lead to long-term harm, including suicidal ideation and work burnout [2], advocacy information needs to be disseminated to HCWs, policymakers, and stakeholders to boost policy and community support toward HCWs during the pandemic.

Corroborating a previous study, the prevalence of PTSS was higher among those reporting separation or divorced marital status. It is plausible that separation or divorced HCWs were at higher risk as they could not be supported by a spouse [6]. Prevention and treatment interventions are greatly warranted and may pay attention to unmarried HCWs. Some studies reported that nurses were at greater risk of PTSS during the COVID-19 pandemic, plausibly because of their closer contact with patients [6,18]. Our findings, however, found that doctors and nurses were equally vulnerable to PTSS, as the difference in their prevalence of PTSS was non-significant. This finding may suggest that doctors also faced tremendous stress such as under-supply of protective equipment and huge uncertainties in face of a new pandemic and had to carry out high-risk and highly stressful treatment procedures during the unprecedented COVID-19 outbreak in China.

The regression model of the present study explained a sizable proportion of the variance of PTSS and may have practical implications. Corroborating other studies, worries about infection risks, coping strategies (rumination, catastrophizing, and acceptance), and perceived family support were independently associated with PTSS [6,18]. The relationship between worries about COVID-19 infection and depression has been widely reported, whereas its relationship with PTSS is less investigated [37]. It is important to alleviate such worries by disseminating regular and updated information regarding pandemic control from credible sources and providing counseling to HCWs. Unexpectedly, working on the frontline was not significantly associated with PTSS, corroborating a previous study in China [38]. This suggests that both the frontline and second-line HCWs may have worries about risk of infection and face similar threats due to COVID-19. There is news that HCWs who did not work in the isolation wards or fever clinics died from COVID-19, which may have resulted from inadequate precautions and insufficient protection in the early stages of the epidemic [39]. One of the novel findings of this study is that perceived difficulty in family caregiving was independently associated with PTSS. It is of pivotal importance to provide HCWs with both instrumental and emotional support during the pandemic, especially those related to family caretaking. In addition, coping strategies showed significant associations with PTSS and mediated a substantial proportion of the association between COVID-19 work-related stressors and PTSS. The confirmed mediation model provided empirical support for the application of the stress-coping theory in addressing HCWs’ mental health problems during the pandemic. Hence, prevention programs should consider coping skills training to help HCWs relieve work stress and adapt to the pandemic positively.

Interestingly, maladaptive coping (e.g., rumination and catastrophizing) exhibited a much stronger mediation effect than adaptive coping (positive reframing and acceptance) (proportion mediated: 39% vs. 4%). As COVID-19 is a novel infectious disease, characterized by tremendous uncertainties and harms, maladaptive emotion regulations, such as rumination and catastrophizing, were commonly elicited during the pandemic [40,41]. In previous pre-COVID studies, stressful and traumatic events were associated with maladaptive coping [14,15], and maladaptive coping was associated with PTSS [18]. The findings confirm that such is also true for COVID-19. In contrast, adaptive coping strategies had small protective effects against PTSS, potentially resulting from COVID-19. For instance, positive reframing was not significantly associated with PTSS in the regression analysis. The findings corroborate previous studies showing that maladaptive coping strategies had stronger associations with mental illness relative to adaptive coping [42]. It seems that prevention interventions reducing maladaptive coping might be more effective than those enhancing adaptive coping. Future studies are needed to confirm the finding.

Furthermore, the direct and indirect associations between COVID-19 work-related stressors and PTSS were significantly moderated by perceived family support. Perceived family support may enable one’s adaptive capabilities to face challenges and overcome adversity [21]. The findings suggested that perceived family support enhanced the protective effect of adaptive coping on PTSS and buffered the risk effect of COVID-19 work-related stressors on PTSS. Specifically, the mediation role of adaptive coping in the association between COVID-19 work-related stressors and PTSS was stronger and only significant among HCWs with higher family support, whereas the association between COVID-19 work-related stressors and PTSS was more evident among HCWs with lower perceived family support. Notably, the moderation role of perceived family support in the indirect path via maladaptive coping was also close to the significance level (0.05 < *p* < 0.10). Therefore, the results generally support the stress-coping theory that coping resources would facilitate effective coping and moderate the relationship of stressors, coping, and health outcomes [21,22]. Promotion and training about adaptive coping with the pandemic might be especially effective for HCWs with available coping resources. Programs preventing PTSS should pay extra attention to those who have low family support, such as HCWs who are non-locals or not living with their families.

These findings have important theoretical implications. First, the present study supports the stress-coping theory generally. However, the appraisal part of the stress-coping theory has not been tested in this study and requires future investigations. Second, the significant direct effect of COVID-19 work-related stressors on PTSS suggests the existence of other potential mechanisms untested in this study. For instance, two empirical studies showed that hardiness and resilience mediated the association between COVID-19-related stress and post-trauma psychopathology among emergency workers and university students [19,20]. Other mediating coping strategies (e.g., active coping and seeking support) between stressors and PTSS may also be important, as the inclusion of the mediators in this study was arbitrary and non-exhaustive. Future studies should test a wider range of potential mediators between COVID-19 work-related stressors and PTSS across countries, populations, and periods, as well as other theories (e.g., conservation of resources theory and general strain theory), to understand HCWs’ mental health problems.

Regarding practical implications, firstly, the high prevalence of PTSS among HCWs signifies the need to implement universal prevention interventions in healthcare settings. Secondary prevention of early and continuous detection of high-risk individuals (e.g., through online self-administered questionnaires) followed by timely counseling and treatment was effective in reducing PTSS [43]. According to our data, related interventions firstly need to reduce the studied COVID-19 work-related stressors that were positively associated with PTSS. Adequate protection equipment should be made available; standard operating procedures and training to prevent nosocomial infections should be implemented to reduce worries about infection. Evidence-based stress reduction programs such as physical exercise, online mindfulness-based stress reduction, and positive psychology interventions may be useful [43]. Second, support should be given to HCWs’ caretaking. For instance, some countries have taken measures to ensure continuity of parental care of children of HCWs at home (e.g., shift work and free childcare center services) [44]. Third, online and offline continuous training, including psychoeducation, should be provided to HCWs. Such training may aim at improving coping strategies (especially reduction of maladaptive coping), removing potential self-blame for giving less care to the family, and empowering confidence in handling future outbreaks. A previous study suggested that psychological training on emotional regulation and positive coping with COVID-19 was positively associated with mental health help-seeking among HCWs during the pandemic [45]. Last but not least, it is important to enhance family support, which may reduce PTSS. For instance, online support groups may be formed to achieve those ends. Time for family and channels for communication with family members must be arranged.

### 4.2. Limitations

This study has several limitations. First, reporting bias about COVID-19 work-related stressors and PTSS due to social desirability may exist. Females overrepresented the sample of doctors (62.9% compared to the Census data of 46.7% in the 2020 China Health Statistical Yearbook). Selection bias might exist if male doctors were less likely to participate in the study and report psychological stress. Second, the cross-sectional study cannot allow for causal inferences. It is plausible that individuals with maladaptive coping (e.g., rumination and catastrophizing) are more likely than others to have excessive worries about infection and perceive higher COVID-19-related stress, whereas those with PTSS may be more inclined to have maladaptive coping. Longitudinal studies are warranted to ascertain the causal relationships between these variables. Third, despite an acceptable response rate, only five hospitals in four provinces in China were involved; generalization to hospitals of other provinces needs caution. Fourth, in the absence of existing tools in the context of the COVID-19 pandemic, some items related to COVID-19 work-related stressors were constructed for this study with reference to previous studies. Family support was measured using a single item. The results of the present study thus need to be confirmed by using refined measurements. Fifth, we only examined a few maladaptive and adaptive coping styles, and the internal reliability of rumination subscale was low. Future studies may consider the full version of CERQ to assess the impact of various coping strategies.

## 5. Conclusions

This study represents an important contribution to understanding the impact and underlying mechanisms of COVID-19 work-related stress on PTSS among HCWs via maladaptive and adaptive coping, which is supportive of the stress-coping theory. The problem of PTSS might affect almost half of the indispensable workforce of HCWs. Early screening, prevention, and treatment are pivotal. Efforts are needed to reduce heavy work-related stress directly through organizational efforts, and indirectly through supportive measures. Attention needs to be paid to improve coping, especially reducing maladaptive coping that might be common during COVID-19 outbreaks. The novel findings of the moderation of the mediations by family support suggest that efforts should be made to foster family support, and possibly other sources of social support. Cross-country and longitudinal studies are required to confirm the findings and to facilitate the design of effective interventions.

### Relevance for Clinical Practice

Findings of how the COVID-19 pandemic impacted HCWs’ mental health shed light on improving mental healthcare for HCWs with traumatic experiences. Relevant health promotion should target enhancement of adaptive coping and especially alleviation of maladaptive coping, strengthening of family support and wider social support systems, and helping HCWs to deal effectively with the unprecedented stress of the pandemic. Maintaining adaptive coping in face of COVID-19-related stress should be particularly effective in HCWs with available family support to combat the negative mental health consequences of the pandemic.

## Figures and Tables

**Figure 1 ijerph-19-06201-f001:**
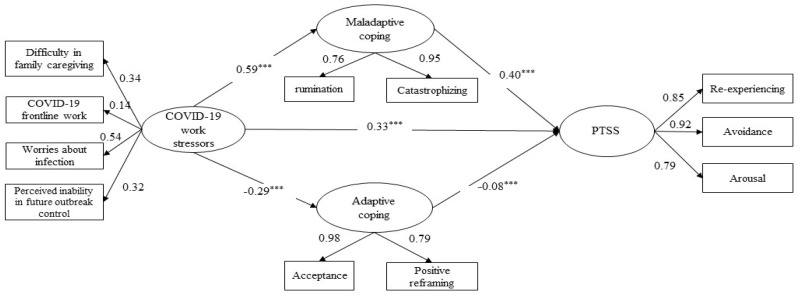
The structural equation modeling of the association between COVID-19 work-related stressors and PTSS via adaptive and maladaptive coping. Note: Latent variables are shown in ovals and observed variables in rectangles. Standardized coefficients are presented (*** *p* < 0.001). For simplicity, the significant background covariates (i.e., marital status) of outcomes and error covariance are not presented. Abbreviations: PTSS, posttraumatic stress symptoms.

**Table 1 ijerph-19-06201-t001:** Description of the study participants’ characteristics (*n* = 1449).

	Total (*n* = 1449)	Doctors (*n* = 423)	Nurses (*n* = 1026)	*p*-Value
Background variables				
Sex				
Male	211 (14.6%)	157 (37.1%)	54 (5.3%)	<0.001
Female	1238 (85.4%)	266 (62.9%)	972 (94.7%)	
Age, mean (SD)	34.1 (9.0)	37.8 (9.1)	32.7 (8.6)	<0.001
Department				
Internal medicine	337 (23.3%)	117 (27.7%)	220 (21.6%)	0.004
Surgery	230 (15.9%)	57 (13.5%)	173 (17.0%)	
Gynecology and obstetrics	80 (5.5%)	21 (5.0%)	59 (5.8%)	
Pediatrics	80 (5.5%)	23 (5.5%)	57 (5.6%)	
Infectious diseases	100 (6.9%)	23 (5.5%)	77 (7.6%)	
Emergency	50 (3.5%)	24 (5.7%)	26 (2.6%)	
Intensive Care Unit	83 (5.7%)	17 (4.0%)	66 (6.5%)	
Others	481 (33.2%)	140 (33.2%)	341 (33.5%)	
Job seniority rank				
Junior	828 (57.1%)	159 (37.6%)	669 (65.2%)	<0.001
Middle	426 (29.4%)	140 (33.1%)	286 (27.9%)	
Vice-senior	129 (8.9%)	78 (18.4%)	51 (5.0%)	
Senior	38 (2.6%)	36 (8.5%)	2 (0.2%)	
Others (e.g., uncertain)	28 (1.9%)	10 (2.4%)	18 (1.8%)	
Marital status				
Single	342 (23.6%)	68 (16.1%)	274 (26.7%)	<0.001
Married/cohabited	1059 (73.1%)	340 (80.4%)	719 (70.1%)	
Others	48 (3.3%)	15 (3.5%)	33 (3.2%)	
Education level				
Junior college or below	398 (27.5%)	34 (8.0%)	364 (35.5%)	<0.001
Bachelor’s degree	990 (68.3%)	333 (78.7%)	657 (64.0%)	
Postgraduate degree	61 (4.2%)	56 (13.2%)	5 (0.5%)	
COVID-19 work-related stressors				
Engagement in COVID-19 frontline work				
No	718 (49.6%)	205 (48.5%)	513 (50.0%)	0.595
Yes	731 (50.4%)	218 (51.5%)	513 (50.0%)	
Worries about being infected with COVID-19, mean (SD)	5.5 (1.8)	5.6 (1.7)	5.5 (1.8)	0.824
Perceived difficulty in family caregiving during the COVID-19 outbreak period				
No	668 (46.1%)	162 (38.3%)	506 (49.3%)	<0.001
Yes	781 (53.9%)	261 (61.7%)	520 (50.7%)	
Perceived inability in handling work during prospective COVID-19 outbreaks, mean (SD)	3.8 (1.5)	4.0 (1.5)	3.8 (1.4)	0.002
Coping, mean (SD)				
Rumination	5.3 (1.5)	5.2 (1.5)	5.4 (1.5)	0.054
Catastrophizing	4.5 (1.5)	4.4 (1.4)	4.5 (1.5)	0.166
Positive reframing	7.2 (1.6)	7.2 (1.6)	7.2 (1.6)	0.985
Acceptance	7.5 (1.6)	7.5 (1.5)	7.5 (1.6)	0.893
Perceived family support				
Low	446 (25.7%)	110 (26.0%)	262 (25.5%)	0.853
High	1287 (74.3%)	313 (74.0%)	764 (74.5%)	
Posttraumatic stress symptoms				
Total mean score (SD)	12.0 (9.9)	12.5 (10.0)	11.8 (9.8)	0.217
Prevalence, n (%) (≥13)	609 (42.0%)	189 (44.7%)	420 (40.9%)	0.189

Note: Data are presented as *n* (%) unless specified. Abbreviation: SD, standard deviation.

**Table 2 ijerph-19-06201-t002:** Multivariable associations between the studied variables and PTSS.

	Posttraumatic Stress Symptoms		
	Standardized Beta	Unstandardized B (95% CI)	*p*-Value
Background variables			
Sex (male vs. females)	−0.00	−0.06 (−1.42, 1.29)	0.928
Age	0.02	0.02 (−0.06, 0.10)	0.606
Department			
Internal medicine	0 (Ref)	0 (Ref)	
Surgery	−0.02	−0.55 (−1.99, 0.90)	0.459
Gynecology and obstetrics	0.01	0.32 (−1.66, 2.31)	0.750
Pediatrics	−0.03	−1.46 (−3.55, 0.63)	0.169
Infectious diseases	−0.03	−1.33 (−3.17, 0.52)	0.159
Emergency	−0.01	−0.32 (−2.75, 2.11)	0.797
Intensive Care Unit	−0.01	−0.27 (−2.24, 1.70)	0.789
Others	−0.01	−0.27 (−1.45, 0.92)	0.661
Job seniority rank			
Junior	0 (Ref)	0 (Ref)	
Middle	−0.01	−0.31 (−1.57, 0.94)	0.627
Vice-senior	0.04	1.40 (−0.73, 3.53)	0.197
Senior	0.00	0.06 (−3.32, 3.45)	0.970
Others (e.g., uncertain)	0.01	0.56 (−2.46, 3.59)	0.714
Type of profession (doctors vs. nurses)	−0.02	−0.43 (−1.63, 0.76)	0.478
Marital status			
Single	0 (Ref)	0 (Ref)	
Married/cohabited	0.03	0.57 (−0.60, 1.74)	0.339
Others (e.g., separated/divorced, widowed)	0.07	3.69 (1.06, 6.32)	**0.006**
Education level			
Junior college or below	0 (Ref)	0 (Ref)	
Bachelor’s degree	0.03	0.61 (−0.45, 1.66)	0.262
Postgraduate degree	−0.01	−0.50 (−3.06, 2.05)	0.700
Hospitals			
Hospital 1	0 (Ref)	0 (Ref)	
Hospital 2	0.01	0.19 (−0.94, 1.32)	0.738
Hospital 3	0.01	0.54 (−1.28, 2.37)	0.559
Hospital 4	−0.03	−0.89 (−2.41, 0.64)	0.254
Hospital 5	0.00	0.00 (−1.67, 1.68)	0.996
COVID-19 work-related stressors			
Engagement in COVID-19 frontline work	0.03	0.66 (−0.23, 1.55)	0.144
Worries about being infected with COVID-19	0.12	0.65 (0.39, 0.90)	**<0.001**
Perceived difficulty in family caregiving during the COVID-19 outbreak period	0.08	1.52 (0.62, 2.43)	**0.001**
Perceived inability in handling work during prospective COVID-19 outbreaks	0.03	0.20 (−0.13, 0.53)	0.239
Coping			
Rumination	0.18	1.21 (0.78, 1.65)	**<0.001**
Catastrophizing	0.33	2.18 (1.73, 2.62)	**<0.001**
Positive reframing	0.01	0.05 (−0.39, 0.50)	0.810
Acceptance	−0.15	−0.91 (−1.33, −0.49)	**<0.001**
Perceived family support	−0.07	−1.50 (−2.54, −0.46)	**0.005**

Note: All independent variables were entered into the multivariable linear regression model. Bold values denote statistical significance at the *p* < 0.05 level.

**Table 3 ijerph-19-06201-t003:** Correlations between the studied variables.

Measure	1	2	3	4	5	6	7	8	9
1. Engagement in COVID-19 frontline work	-								
2. Worries about being infected with COVID-19	0.11 **								
3. Perceived difficulty in family caregiving during the COVID-19 outbreak period	0.14 **	0.20 **							
4. Perceived inability in handling work during prospective COVID-19 outbreaks	−0.06 *	0.14 **	0.09 **						
5. Rumination	0.15 **	0.27 **	0.15 **	−0.02					
6. Catastrophizing	0.06 *	0.34 **	0.15 **	0.16 **	0.72 **				
7. Positive reframing	0.08 **	−0.10 **	−0.03	−0.40 **	0.13 **	−0.11 **			
8. Acceptance	0.07 *	−0.09 **	−0.05 *	−0.36 **	0.06 *	−0.17 **	0.79 **		
9. Perceived family support	0.06 *	−0.03	−0.02	−0.28 **	−0.02	−0.17 **	0.30 **	0.29 **	
10. PTSS	0.09 **	0.31 **	0.21 **	0.17 **	0.45 **	0.54 **	−0.16 **	−0.23 **	−0.17 **

* *p* < 0.05, ** *p* < 0.01.

**Table 4 ijerph-19-06201-t004:** Results of multi-group structural equation modeling by levels of perceived family support.

	Χ^2^	df	CFI	RMSEA	Δχ^2^	Δdf	*p*-Value
Unconstrained model (Model 1)	622.14	92	0.923	0.055	-	-	-
Constrained model							
Model 2 (COVID-19 work-related stressors to maladaptive coping)	625.71	91	0.923	0.059	3.56	1	0.059
Model 3 (COVID-19 work-related stressors to adaptive coping)	719.08	91	0.909	0.064	96.94	1	**<0.001**
Model 4 (maladaptive coping to PTSS)	625.36	91	0.923	0.059	3.21	1	0.073
Model 5 (adaptive coping to PTSS)	626.44	91	0.923	0.059	4.29	1	**0.038**
Model 6 (COVID-19 work-related stressors to PTSS)	628.47	91	0.923	0.059	6.33	1	**0.012**

Note: Abbreviations: PTSS, posttraumatic stress symptoms. The specific path in parentheses was constrained to be equal across high and low perceived family support groups for each tested model. Bold values denote statistical significance at the *p* < 0.05 level.

## Data Availability

The data presented in this study are available on request from the corresponding author.

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
