# Peer review of "Associations between COVID-19 Work-Related Stressors and Posttraumatic Stress Symptoms among Chinese Doctors and Nurses: Application of Stress-Coping Theory"

_ijerph, 2022, doi:10.3390/ijerph19106201_

Round 1

Reviewer 1 Report

The work I read seemed to be precise in the exposition of the concepts. The premise is a bit long but it doesn't seem like a major flaw. What is not clear is what the authors mean by the phrase "put under control" referring to the health emergency that has just ended. This sentence is repeated several times in the manuscript but it is not clear what are the elements that allow us to consider COVID-19 under control since the pandemic has spread worldwide and just while in other countries there were critical situations in terms of virus spread. Also, as mentioned above, there are minor typos and spelling errors that can be fixed by carefully rereading the manuscript.

Best Regards

Author Response

Response to Reviewer

Reviewer: 1

The work I read seemed to be precise in the exposition of the concepts. The premise is a bit long but it doesn't seem like a major flaw. What is not clear is what the authors mean by the phrase "put under control" referring to the health emergency that has just ended. This sentence is repeated several times in the manuscript but it is not clear what are the elements that allow us to consider COVID-19 under control since the pandemic has spread worldwide and just while in other countries there were critical situations in terms of virus spread. Also, as mentioned above, there are minor typos and spelling errors that can be fixed by carefully rereading the manuscript.

Response: Thanks for the comment. After the initial outbreak of COVID-19 in February 2020 in China, the cases of COVID-19 had dropped dramatically since April 2020. Specifically, the national daily number of newly reported COVID-19 cases in China peaked at 15,512 on February 12th 2020 and then declined onward and dropped to less than 10 per day on April 22nd 2020, and remained at a very low level ever since (2020). The present study was conducted from October to November 2020, which was about six months since COVID-19 was “put under control” in China, when hospitals in China were providing normal services to the public (Wu, Chen et al. 2020). We think the timing of the study investigation is appropriate, as it would be more accurate to assess PTSS after the trauma (i.e., the COVID-19 outbreak). We have specified in the introduction, “To monitor the longer-term impact of COVID-19, it is important to investigate PTSS among HCWs during a “post-outbreak period”, i.e., when the newly detected COVID-19 cases have completely or largely been subsided. Under those circumstances, individuals showing PTSS may develop lasting problems and would require interventions (Carty et al., 2006). China is one of the few countries that offer such a unique context for investigating PTSS among HCW during a “post-outbreak period”. While many countries are still reporting a large number of new cases, the national daily number of newly reported COVID-19 cases in China peaked at 15,512 on February 12th 2020 and then declined on-ward and dropped to less than 10 per day on April 22nd 2020, and remained at a very low level ever since (2020). The present study was conducted from October to November 2020, which was about six months since COVID-19 was “put under control” in China, when hospitals in China were providing normal services to the public (Wu et al., 2020). The assessment of PTSS among HCWs during that time period was appropriate.” (page 2)

Reviewer 2 Report

The manuscript by She et al., aims to examine the prevalence of post-traumatic stress symptoms (PTSS) in a population of Chinese health care workers and to determine whether maladaptive and adaptive coping strategies, and perceived support from family members were associated with PTSS. They undertook this study via a voluntary online survey which targeted five hospitals across 4 health districts. They found that PTSS was present in 42% of health care workers assessed and that maladaptive coping had stronger mediation effect that adaptive coping on PTSS. Thus, prevention of maladaptive coping rather that enhancing adaptive coping might be more effective at preventing PTSS. In addition, the found that higher levels of perceived family support enhanced adaptive coping strategies thus lower the risk of PTSS. Individuals who were in a single living situation were more likely to develop maladaptive coping strategies and PTSS. These findings highlight at-risk health care worker communities and indicate that intervention by community-building and resilience exercises may help reduce PTSS in health care workers.

Overall, I commend the authors on a well-executed study. They have clearly outlined the limitations of the data set collected and have performed thorough analysis and modelling that has yielded important insights into PTSS in health care workers in the current COVID-19 pandemic.

Abstract

Sentence beginning “multi-group analyses…” at line 30 – hard to interpret. Consider dividing into two sentences to make meaning clearer.

Methods

Concern that sampling has been labelled ‘convenient’. Suggest the authors re-word their sampling design.

Cronbach’s alpha 0.66 for rumination is under the usual acceptable limit. While often unavoidable this needs to be addressed in the limitations.

Results

Mostly female respondents – bias in data set. Men do not want to/are less likely to report stress?

Table 1 formatting: subcategories more clearly nested underneath their parent category. At present they are difficult to differentiate.

Interesting that PTSS does not appear to be more prevalent in those who have/have not worked directly with COVID patients. Any similar reports in published literature?

Author Response

Reviewer: 2

The manuscript by She et al., aims to examine the prevalence of post-traumatic stress symptoms (PTSS) in a population of Chinese health care workers and to determine whether maladaptive and adaptive coping strategies, and perceived support from family members were associated with PTSS. They undertook this study via a voluntary online survey which targeted five hospitals across 4 health districts. They found that PTSS was present in 42% of health care workers assessed and that maladaptive coping had stronger mediation effect that adaptive coping on PTSS. Thus, prevention of maladaptive coping rather that enhancing adaptive coping might be more effective at preventing PTSS. In addition, the found that higher levels of perceived family support enhanced adaptive coping strategies thus lower the risk of PTSS. Individuals who were in a single living situation were more likely to develop maladaptive coping strategies and PTSS. These findings highlight at-risk health care worker communities and indicate that intervention by community-building and resilience exercises may help reduce PTSS in health care workers.

Overall, I commend the authors on a well-executed study. They have clearly outlined the limitations of the data set collected and have performed thorough analysis and modelling that has yielded important insights into PTSS in health care workers in the current COVID-19 pandemic.

Abstract

Sentence beginning “multi-group analyses…” at line 30 – hard to interpret. Consider dividing into two sentences to make meaning clearer.

Response: Thanks for the comment. We have revised the sentence, “Multi-group analyses results showed that perceived family support tended to intensify the associations between COVID-19 work-related stressors and adaptive coping and between adaptive coping and PTSS. While perceived family support attenuated the positive association between COVID-19 work-related stressors and PTSS.” (page 1)

Methods

Concern that sampling has been labelled ‘convenient’. Suggest the authors re-word their sampling design.

Response: Thanks for the comment. We have clarified the sampling procedure, “An anonymous cross-sectional survey was conducted from October to November 2020. Five hospitals in four provinces (Zhejiang, Ningxia, Guangxi, and Yunnan) in mainland China were selected via existing collaboration network. The four provinces were geographically (east, north central, south, and southwest) and socioeconomically (levels of gross domestic product: top, about average, below average, and low) representative of mainland China to some extent.” In order to avoid any potential confusion, we have deleted the label of “convenience sampling”. (page 4)

Cronbach’s alpha 0.66 for rumination is under the usual acceptable limit. While often unavoidable this needs to be addressed in the limitations.

Response: Thanks for the comment. We have acknowledged it in the limitation, “Fifth, we only examined a few maladaptive and adaptive coping styles and the internal reliability of rumination subscale is low. Future studies may consider the full version of CERQ to assess the impact of various coping strategies.” (page 14)

Results

Mostly female respondents – bias in data set. Men do not want to/are less likely to report stress?

Response: Thanks for the comment. According to the 2020 China Health Statistical Yearbook, 97.2% of registered nurses and 47% of physicians are female. As we are not able to get the response rate by sex, we have discussed the possibility of response bias in the limitation, “Females overrepresented the sample of doctors (62.9% compared to the Census data of 46.7% in the 2020 China Health Statistical Yearbook). Selection bias might exist if male doctors were less likely to participate in the study and report psychological stress.” (page 13)

Table 1 formatting: subcategories more clearly nested underneath their parent category. At present they are difficult to differentiate.

Response: Thanks for the comment. I think the format of Table 1 followed the guideline and template of this journal. I have now underlined the subcategories to make it clear for presentation. (page 7)

Interesting that PTSS does not appear to be more prevalent in those who have/have not worked directly with COVID patients. Any similar reports in published literature?

Response: Thanks for the comment. We have added the discussion regarding the result, “Unexpectedly, working on the frontline was not significantly associated with PTSS, corroborating a previous study in China (Zhang, Shi et al. 2020). This suggests that both the frontline and second-line HCWs may have worries about risk of infection and face similar threats due to COVID-19. There is news that HCWs who did not work in the isolation wards or fever clinics died from COVID-19, which may have resulted from inadequate precautions and insufficient protection in the early stages of the epidemic (Zhan, Qin et al. 2020).” (page 12)

Reviewer 3 Report

The authors has surveyed posttraumatic stress reactions and related factors among Chinese health care workers 6month post the first pandemic of COVID-19. They have elucidated that perceived family support tended to mitigate the post-traumatic stress of fighting COVID-19. The manuscript has importance in the field of disaster psychiatry. The manuscript is well written and I only have several minor comments as described below.   p2  line 46 "Acute stress disorder" may be rephrased to "Acute stress reactions (ASR)" throughout the manuscript, to coincide with the  use of PTSS.   p4 line 185, p5 line 213 COVID-19 work-related stressors and perceived family support, I assume that the questionnaires are originally created by the authors,  please describe that the questions were original.

Author Response

Reviewer: 3

The authors has surveyed posttraumatic stress reactions and related factors among Chinese health care workers 6month post the first pandemic of COVID-19. They have elucidated that perceived family support tended to mitigate the post-traumatic stress of fighting COVID-19. The manuscript has importance in the field of disaster psychiatry. The manuscript is well written and I only have several minor comments as described below.   p2  line 46 "Acute stress disorder" may be rephrased to "Acute stress reactions (ASR)" throughout the manuscript, to coincide with the  use of PTSS.   p4 line 185, p5 line 213 COVID-19 work-related stressors and perceived family support, I assume that the questionnaires are originally created by the authors,  please describe that the questions were original.

Response: Thanks for the comment. We have revised the “acute stress disorder” to “acute stress reactions”. For the measurements of COVID-19 work-related stressors and family support, we have mentioned that they were original items. We also acknowledged it in the limitation, “Fourth, in the absence of existing tools in the context of the COVID-19 pandemic, some items related to COVID-19 work-related stressors were constructed for this study with reference to previous studies. Family support was measured using a single item. The results of the present study thus need to be confirmed by using refined measurements.” (page 14)

Reviewer 4 Report

The article is very interesting because of the chosen objective and the research methods used.
I suggest the authors change the title. The title of the article should refer to the research group, i.e. medical personnel working in covidien wards.
I have no other objections to the article.

Author Response

Reviewer: 4

The article is very interesting because of the chosen objective and the research methods used.
I suggest the authors change the title. The title of the article should refer to the research group, i.e. medical personnel working in covidien wards.
I have no other objections to the article.

Response: Thanks for the positive comment. We should clarify that only part of investigated HCWs had direct contact with the COVID-19 patients. Therefore, it might not be appropriate to add covidien wards in the title.